# Comparison of altruistic, egoistic, and scientific appeals in the recruitment of politicians to a survey panel

Lauri Rapeli[1]*, Kim Backström[2]

1 The Social Science Research Institute, Åbo Akademi University, Åbo, Finland, 2 The Social Science Research Institute, Åbo Akademi University, Vasa, Finland

* lauri.rapeli@abo.fi

## Abstract

Low survey participation is a persistent problem for all survey researchers. Unlike citizens, politicians are seldom targeted for surveys. Politicians often are not prepared to participate, which leads to a lack of understanding regarding their willingness to collaborate in surveys. This study presents a rare survey recruitment experiment where 7,397 local-level politicians from Finland were randomly assigned to groups that received by e-mail 1) an altruistic or 2) an egoistic appeal or 3) an appeal referring to benefits for science when they were invited to participate in the Finnish Politician Panel. Overall, we find no differences in collaboration rates, suggesting that in politician samples the altruism-egoism framework is largely irrelevant and that for maximizing participation rates among politicians, survey researchers should look into alternative methods. However, politicians who are older, live in urban municipalities and represent the political left are more likely than their counterparts to volunteer as panel participants.

## Introduction

Getting people to participate in surveys is an ever-present dilemma for any organization that conducts surveys. Declining response rates [(e.g. 1)] have in recent years made the issue even more salient, because respondent recruitment has become more critical for the success of any survey project. The issue is even more critical for panel studies, which require much more participant commitment than participation in individual surveys.

Compared to samples of citizens, politicians are less often the target population for survey recruitment efforts. Citizens are much more readily available through various survey companies, research panels and other sources, whereas politicians are harder to contact and they may not be quite as prepared to expose themselves to researchers' questionnaires. Nevertheless, in recent years, researchers have initiated several large-scale studies, especially among local-level politicians, such as the Civic

**Data availability statement:** The data and replication code are deposited at https://osf.io/zy27n/files/osfstorage.

**Funding:** This study has received funding from The Research Council of Finland grants Research 327997, 345714 and 361424.

**Competing interests:** The authors have declared that no competing interests exist.

Pulse in the US, the European Panel of Local Officials (https://localgovernancelab.com/) and the University of Tokyo survey of Japanese legislators (https://www.masaki.j.u-tokyo.ac.jp/utas/utasindex_en.html). While these much needed efforts provide a wealth of new evidence regarding the attitudes and behaviors of elected local-level politicians, so far researchers have not had much focus on how to approach politicians when recruiting them to participate in surveys and panel studies.

Previous studies show that getting politicians to participate in academic surveys is not always an impossible task, although the limited number of politicians makes successful recruitment even more important than in the case of citizens [(e.g. 2, 3)]. However, politicians typically receive numerous requests to attend meetings, give interviews, make public appearances and also to participate in surveys, which makes them a challenging group to recruit for research purposes and whose response rates are often lower than among citizen samples [4,5]. In addition to being in 'high demand', Walgrave and Joly [5] argue that politicians dislike being considered just another 'number' in anonymous survey data as they would much rather be seen as unique individuals with a story to tell and an agenda they wish to promote.

Consequently, survey participation may seem especially unappealing to politicians, compared with ordinary citizens. While the handful of studies, which have analyzed survey recruitment of elected politicians, have focused on one-off surveys, our study contributes to existing literature by examining recruitment to a panel study. Volunteering to become part of a panel that receives a couple of surveys per year requires a considerably higher degree of commitment by the invited politicians and makes it even more important for the researchers to have the right recruitment approach.

This study contributes to the scarce literature by examining politicians' willingness to cooperate in a survey panel through a theoretically motivated experimental design. We randomize politicians to three groups, each receiving a different version of the invitation. The *altruistic* message refers to participating to improve how democracy works, the *egoistic* appeal emphasizes the opportunity to voice one's personal views and the *scientific altruistic* message is an appeal to help advance research by participating. The sample consists of 7,397 elected local-level politicians from Finland, who were invited to participate in the Finnish Politician Panel. Running since 2021, the panel has 1,562 participants after the recruitment round in question.

We study the impact of invitation framing for pragmatic reasons. In their role as elected policymakers who serve the interests of the public, accepting – or expecting – money for advancing research on democracy and perhaps their own political accountability does not seem suitable. This is particularly the case in many countries, such as Finland, where research is primarily publicly funded. Instead of monetary incentives, which may be feasible for attracting ordinary citizens to participate in surveys, but not equally appropriate when approaching democratically elected politicians, it is necessary to investigate other ways to improve recruitment success, such as whether different framings could affect politicians' willingness to participate in a survey panel. We examine the possibility that politicians could be persuaded by appealing to their altruistic sense of responsibility to contribute to research about how democracy works, due to their position as elected representatives of the people.

Alternatively, a more egoistic appeal suggests that by becoming a panel participant, the politicians would have an opportunity to make their opinions heard. If effective, simple appeals to participate would be an easy and cost-free method for survey researchers to increase cooperation rates in politician samples.

There are no statistically significant differences across the different appeals. However, we find that women, older, leftist, representing a bilingual or urban municipality are more likely to comply with the panel invitation than their counterparts. Still, we find some heterogeneous effects: egoistic appeals are more effective for politicians representing suburban municipalities, and altruistic appeals are less effective for politicians representing bilingual municipalities with a Swedish majority. While we cannot rule out the possibility that other types of appeals can be more effective, we interpret the findings as suggesting that researchers need to look beyond the framing of invitations in order to maximize recruitment rates among politicians. The perceived trustworthiness and reputation of the inviting organization are potentially more important factors behind politicians' willingness to commit to a survey (panel).

## Survey recruitment strategies

Successful survey recruitment presents an enduring dilemma for survey researchers who struggle with poor cooperation rates and representativeness issues. With politicians, who are not only busy, but typically also receive a much higher number of requests from the public, stakeholders, researchers, and so on than the average person, the recruitment strategy is a particularly important matter. From the perspective of tailoring [6], and its theoretical continuation, the leverage-saliency theory [7], the recruitment of politicians could be more effective if they find the recruitment message relevant and appealing, which nudges them just enough toward accepting the invitation. For survey researchers, messages offer an opportunity to increase recruitment and response rates without any costs. Still, there is a relative lack of proper empirical testing, as, for example, evidenced by the scarcity of studies in the review by Sammut et al. [8].

The practical issue is how can an invitation be tailored so that it has enough saliency for politicians to cooperate? Fundamentally, we draw on the central tenet of social conformity theory, according to which people's behaviors and actions depend on what they assume is expected of them because of their social positions and role in any situation [(see e.g. 9, 10)]. Thus, performing in their role as an elected politician when they receive the invitation, we expect the politicians to be persuaded by appeals that refer to what is perhaps societally expected behavior in this particular situation. Consequently, we assume that politicians may feel that demonstrating a sense of civic duty is something that conforms to social expectations. Therefore, the mechanism, which we expect to trigger a response in the respondents as they read the invitation with the altruistic appeals, is the feeling that politicians' are expected to display a sense of duty towards the good of society. Additionally, research has shown that altruism and a sense of civic duty are strong predictors of political engagement [11,12] and local-level politicians are extraordinarily highly engaged individuals. Research has also shown that a motivating factor in becoming a politician in the first place is civic duty, i.e., contributing (altruistically) to society [13].

On the other hand, politicians' behavior is likely to also be affected by other factors besides a pursuit of social conformity. They are accustomed to competing for attention to survive in electoral competition. Consequently, it also seems plausible that politicians' reactions to survey invitations could be affected by more selfish considerations about how to make themselves heard, rather than simply being considered an anonymous data entry (see also 5). In this case, the mechanism assumed to be at play is the more self-centered need to seek visibility and focus, rather than the need to conform to duty-based role expectations.

From this point of departure, we roughly follow the experimental design of Pedersen and Nielsen [14], who studied the impact of egoistic versus altruistic appeals. An egoistic appeal presents participating in a survey from a respondent-centered viewpoint, either as an opportunity for the respondent to voice their opinion through survey responses or by emphasizing that the respondent has specifically been selected to participate in the survey. An altruistic appeal refers to contributing to the 'common good' or 'public interest'.

Findings regarding the effectiveness of egoistic and altruistic messages are inconclusive. Pedersen and Nielsen report a higher response rate with the egoistic appeal but not with the altruistic one. The non-effect of the altruistic appeal is in line with Hjortskov et al. [15], who found that appeals to 'public service' had no impact on response rates. Brosnan et al. [16], however, found that stated 'benefits to others' from responding to a survey increased cooperation rates. In the current analysis, the target group consists exclusively of politicians, not private citizens. With reference to social conformity theory, politicians seem particularly vulnerable to social desirability bias or concern for 'self-presentation' than the general public [17]. This could make them less persuadable by egoistic messages and more likely to respond positively to altruistic appeals. In addition to a general altruistic appeal, we include a more specific version called the scientific appeal. It emphasizes the importance of volunteering as a panel member to contribute to the advancement of science. Although politicians may hold different opinions regarding science, for most of them, it seems plausible that an appeal to contribute to science triggers a norm-conforming behavioral response that leads them to accept the invitation. Additionally, in line with the saliency theory, the scientific treatment message expects to increase the sense of relevance of participating, which could be particularly effective among politicians who supposedly wish to contribute to the common good. Politicians on the political left could be particularly likely to be persuaded by the scientific appeal, because they tend to have more faith in science [18]. However, the saliency of the scientific benefits of participating could plausibly depend on ideological leaning. In recent years, climate change and Covid-19 have become strongly contentious issues in terms of attitudes towards science and conservatives display significantly less trust in science than liberals and/or left-leaners [18–20]. Distrust of science is also associated with behavioral outcomes, such as lower support for science-based policy and for scientifically grounded recommendations [19]. Consequently, politicians in the political left could be more likely be persuaded by the scientific appeal, because they tend to have more faith in science and also behave accordingly [18].

We test the following three hypotheses:

$H_1$: The societal benefit altruistic appeal will lead to the highest overall acceptance rates to participate in the survey panel

$H_2$: Compared with the societal benefit altruistic appeal, the scientific benefit version of the altruistic appeal will lead to higher acceptance rates among leftist politicians, but not among rightist politicians

$H_3$: The egoistic appeal leads to the lowest overall acceptance rates

## Materials and methods

There are 8,859 elected representatives in the municipal councils in mainland Finland, with Åland Islands excluded, according to the official statistics provided by the Finnish Ministry of Justice, accessed at www.vaalit.fi on Dec 5th 2023. From these, 1,043 were already participants in the Politician Panel and these individuals were excluded from the target population in the experiment. For the remaining 7,816 representatives, personal e-mail addresses were gathered through the municipalities' websites. When the e-mail addresses were not listed on the website, the relevant public official in the municipality was contacted with a request to obtain them. In some cases, the municipality for some reason declined to provide the e-mail addresses, resulting in 419 (5.4 percent) representatives being excluded due to lack of access. Hence, the experiment covered 94.6 percent of the target population. The data gathering took place between June 4 and September 18 2024. According to the statement by the Åbo Akademi University Board of Research Ethics from April 19 2024, the experiment did not require a specific ethical review. The invitees were asked to give their informed consent by accepting the data protection statement of the study and by choosing either the option to continue to the recruitment survey or the option to discontinue. Those who continued to the survey, agreed to the following (translation by the authors): "I have read the data protection statement of the Politician Panel. I approve it and give my informed consent to participate in the Politician Panel." The choice was recorded for each invitee.

An a priori sensitivity test showed that the sample size would be enough to observe an effect size of $f = 0.0448$, while a previous similar study on a general population sample found a 2.18%-units increase in cooperation rate with an effect size

of f = 0.0431. Based on this, we should be able to detect small effects of the treatment. See the preregistration for more details.

The invitation explained that the recipient has been invited to the panel, because (s)he is an elected representative in a municipal council. In the current case, the respondents were targeted due to them holding public office, not as private citizens, and their contact information was obtained from public sources. To obtain informed consent, the respondents were first asked to accept the data protection statement in order to be able to respond to the recruitment survey. They were also informed that if they did not accept the statement, they would not be able to answer any questions or become part of the panel. The data protection statement includes all information about the panel, its administration, data management and privacy issues. After giving consent and accepting the statement, the respondent was taken to a short survey asking about background characteristics (such as age and education).

In Finland, local-level politicians, unlike members of parliament, do not have personal assistants. As with any web-based survey, there is always a risk that the person to whom the survey invitation is intended, delegates the responding to someone else. However, in this particular case, the risk is no greater than for a standard citizen sample. One way of estimating whether it is the politicians themselves who respond to surveys sent to the panel or not is by looking at how consistently they answer items that are likely to be consistent over time. Given that fundamental political attitudes tend to be remarkably consistent even during the life span, we looked at possible fluctuation in the politicians' left-right self-placement across several measurements over a few years' time. We find no meaningful intra-person variation and the self-placements correspond according to expectation with party placement on the same left-right scale. Although this is circumstantial evidence, it provides credibility to the (safe) assumption that it is the politicians themselves who receive the e-mails and respond.

The sample was randomized into three groups, and each was sent a different version of the same invitation. As a randomization check, we compare the sample compositions of the three treatment groups concerning gender, age, language, municipality type, municipality language ratio, and political party classification. We find no statistically significant differences between the groups, indicating successful randomization (see Table A1).

The **egoistic** treatment message emphasized the opportunity to express personal views: *By participating in the panel, you will be able to voice your opinion about democracy in Finnish municipalities and about different societal issues*.

The **altruistic** treatment message, in contrast, emphasized societal benefits: *By participating in the panel, you will benefit the development of democracy in Finnish municipalities, which helps sustain the wellbeing of the Finnish society in the future*.

The **scientific** treatment message highlighted benefits to science: *By participating in the panel, you will contribute to the advancement of scientific research about Finnish democracy, especially on the local level*.

The pre-registration of the experiment is available through the OSF platform (https://doi.org/10.17605/OSF.IO/C3T4X). Although the experiment followed the pre-registration in other respects, it deviated by excluding education as a background characteristic of the population due to unexpected unavailability. Also, the analysis included the mother tongue, municipality type, and language ratio (Finnish monolingual or bilingual with Finnish or Swedish majority) in the municipality as background characteristics of the politicians.

See the Appendix for a closer description of the included variables. Our rationale for including the variables is to improve precision in identifying treatment effects, detect potential biases, and enable us to study heterogeneous treatment effects. As mentioned when discussing the hypotheses, we believe party type affiliation plays a role in the effectiveness of the treatments. Regarding other individual-level demographic variables, we include age due to it regularly being an important predictor of (non-)response in surveys [21]. Gender and language are relevant due to potential differences in treatment effects, due to perceptions of being in the majority or minority locally or nationally. Women are generally underrepresented in politics, meaning they may perceive the treatments differently from men, and likewise, the effect of the treatments may differ for those with a mother tongue other than Finnish. The context-level variables, while potentially

suffering from ecological validity issues, may also provide insights into the context in which the politicians operate and potential differences in the effects of the treatments. There may be differences in resources and time budgets between municipalities of differing urbanicity, while a bilingual context can also have implications for representativeness, e.g., due to differences in political culture. Still, we view our investigation of these potential heterogeneous effects as purely explorative.

## Results

Fig 1 below shows that the recruitment rates for the entire sample and the different treatments vary between 5.5 percent for the *altruistic* treatment message (n = 2466 invited, n = 136 accepted), 5.8 percent for the *egoistic* (n = 2465, n = 144) and 6.2 percent for the *scientific* (n = 2466, n = 153). See Appendix table A2 for an overview of the sample by recruited and non-recruited, Between-group variation is not statistically significant, which leads us to reject $H_1$ and $H_3$. The overall cooperation rate is comparable to previous recruitment rounds to the same panel.

We also examine heterogeneous treatment effects by comparing recruitment rates across different subpopulations. See Figure A1 and Table A4 in the Appendix. We can note some general trends where women, older politicians, and leftist politicians in general are more prone to join the panel. However, there are no significant differences between treatments within the different groups. Still, while marginally insignificant, the egoistic treatment performed somewhat better among politicians in suburban municipalities. Likewise, the egoistic treatment performed somewhat better, although not statistically significantly, among politicians in Swedish-majority bilingual municipalities.

We also use logistic regression and predicted probabilities to examine predictors of recruitment. We test four different models and compare their fit (see Appendix Table A5 and Table A6. The chosen model (model 4) had the best fit, and we further examined its quality using the performance-package [22]. We conclude that the model had a good fit, although it showed some issues with binned residuals, suggesting a slight risk of overpredicting recruitment for some groups (see Figure A2). Across all four models, the overall treatment effects are not significant. As a robustness check, we also tried running the same model using Firth's logistic regression, which can be more suitable when working with unbalanced data [23]. However, we note no substantive differences between the regular MLE logistic regression and Firth's logistic regression (Table A7 in the Appendix), meaning we continue using the MLE logistic regression model.

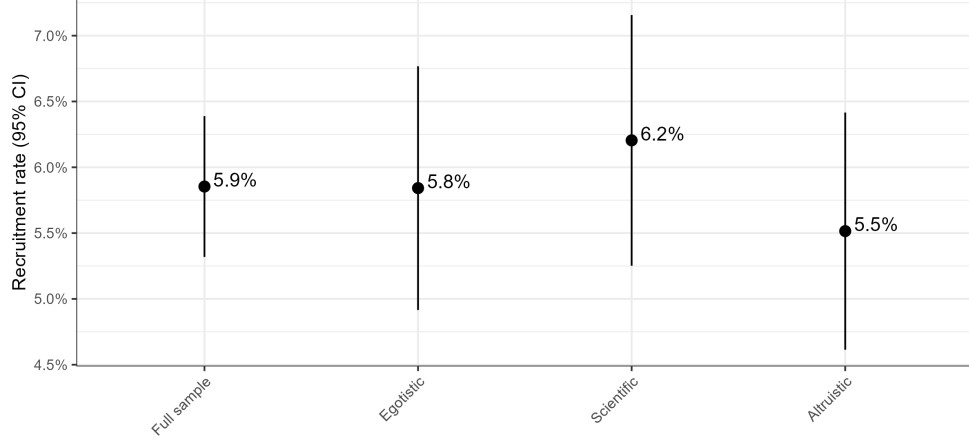

**Fig 1. Recruitment rate (95% CI) for the full sample and by treatment.**

Fig 2 below shows the predicted probabilities of a politician being recruited, calculated across the averages of the following predictors: treatment, gender, age, mother tongue, municipality type, municipality language ratio, and political party type. Furthermore, Table A3 in the Appendix shows the odds ratios of being recruited using the same model.

Four of the predictors are statistically significant, namely gender, age, municipality type (urban vs. rural), municipality language ratio (Finnish monolingual vs. Finnish majority bilingual), and political party type. In terms of odds, men are less likely (36%) to be recruited than women, older politicians are more likely to be recruited (3% for each year older), urban politicians are more likely (37%) to be recruited than rural politicians, politicians in Finnish monolingual municipalities are less likely (42%) to be recruited than politicians in Finnish majority bilingual municipalities, and politicians from leftist parties are more likely (31%) to be recruited than rightist politicians. Finally, an analysis of deviance shows that age is the most important significant predictor of being recruited, followed by gender, municipality language ratio, and political party classification (See Table A8 in the Appendix).

We also look at heterogeneous treatment effects using logistic regressions. See Table A9 for logistic regression models with interactions. Fig 3 demonstrates very weak support for $H_2$. Overall, the scientific appeal has the highest recruitment rate among leftist politicians and the rate is higher than among rightist politicians, but the difference is not statistically significant. Moreover, even among rightist politicians, the scientific appeal has a slightly higher recruitment rate than the societally altruistic appeal.

In Figs 4 and 5, we highlight statistically significant findings outside of the hypotheses, simply as an exploratory approach to identify potential underlying effects that are otherwise hidden by the main effects. Politicians from suburban municipalities are less likely to cooperate when they receive the scientific appeal, compared to politicians from urban municipalities (Fig 4). Potential substantive interpretations for why the scientific appeal would perform worse in a suburban context are challenging. One possibility could be that suburban politicians find academic research to have less of an

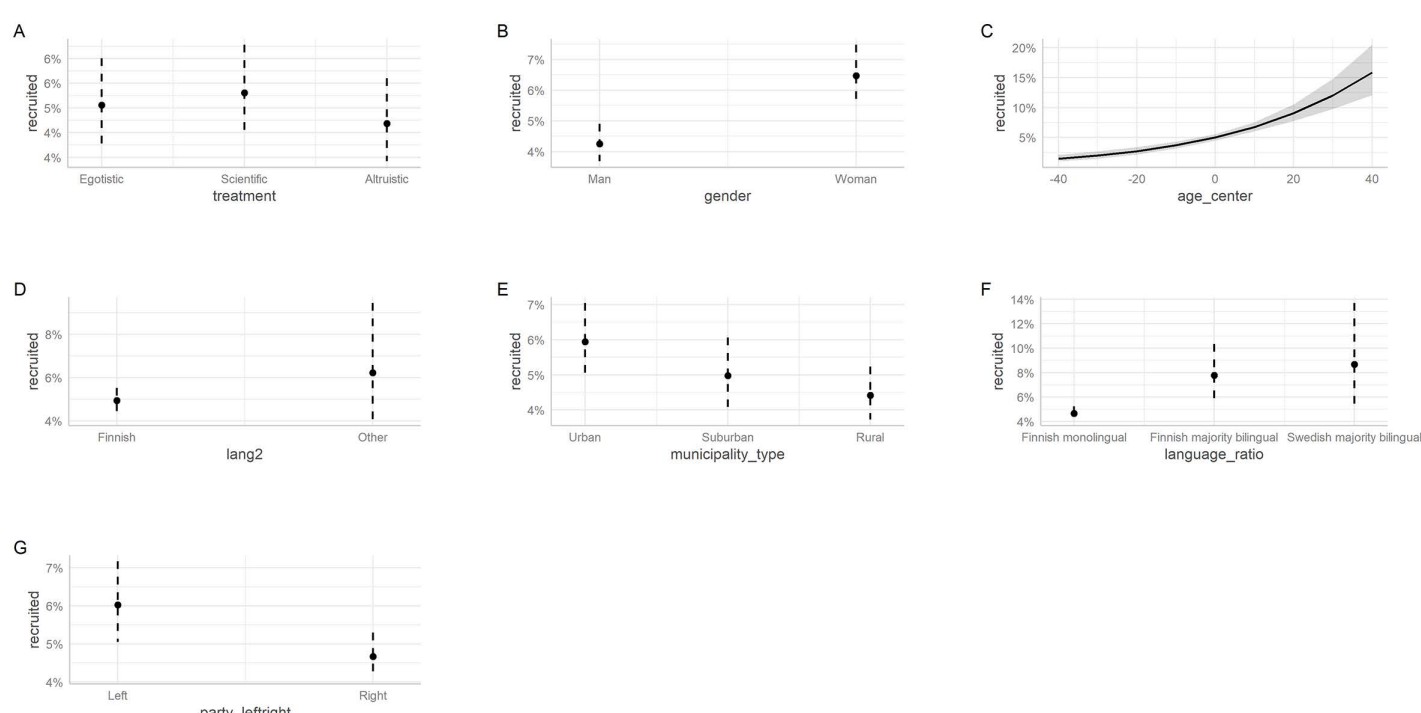

**Fig 2. Predicting recruitment by treatment, gender, age, language, municipality type, language ratio in the municipality, and party type (left or right).**

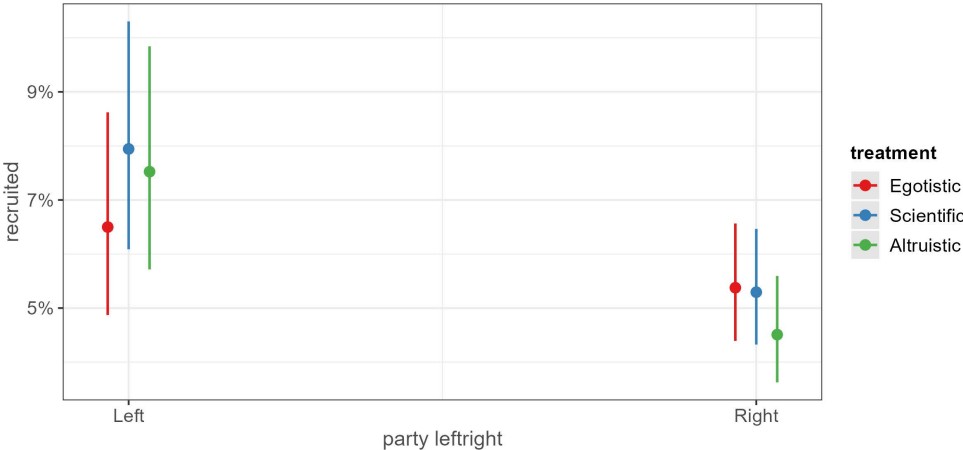

**Fig 3. Interaction party type left-right and treatment.** Note: See Appendix for classification of parties.

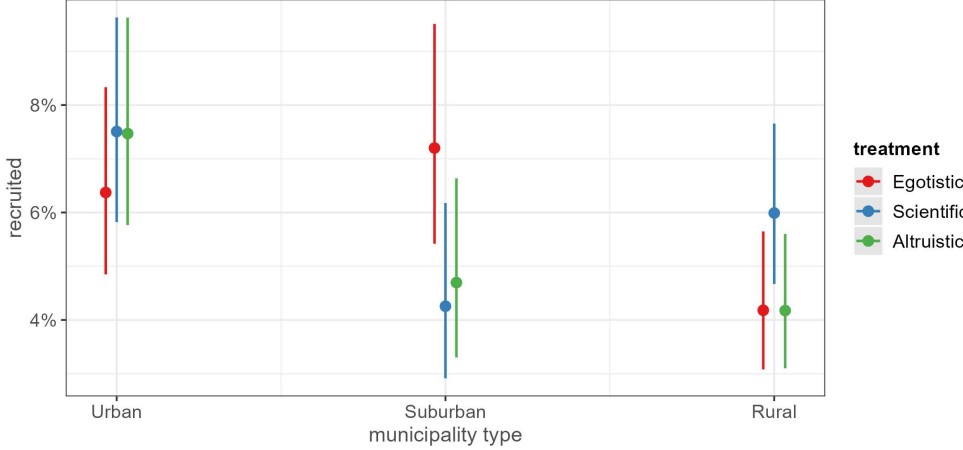

**Fig 4. Interaction municipality type and treatment.**

impact on their lives and communities. Still, this could result from our inability to control for other factors such as educational attainment.

Finnish municipalities also differ in terms of the language balance between Finnish and Swedish speakers. Fig 5 shows the statistically significant and lower recruitment rate in the altruistic treatment group among politicians in municipalities with a Swedish-speaking majority. In these municipalities, the egoistic appeal seems to have been efficient, compared to the altruistic one. Two possible interpretations are that, first, the Swedish-speaking politicians in these communities represent a small ethnic-linguistic minority of about 4.5 percent of the Finnish population. The chance to speak one's mind could be appealing to them because they may feel the need to make themselves heard. Second, the Finnish-speaking politicians living in a minority position in municipalities that are dominated by Swedish-speakers might, in a similar way, be interested in the opportunity to voice their opinion. Therefore, it seems that egoistic appeals may work among sub-populations that find themselves in a minority position.

As a further measure of the representativity of the recruited politicians, we estimate representativity indicators [24]. R-indicators are based on the standard deviation of probabilities of responses of units in the sample. The

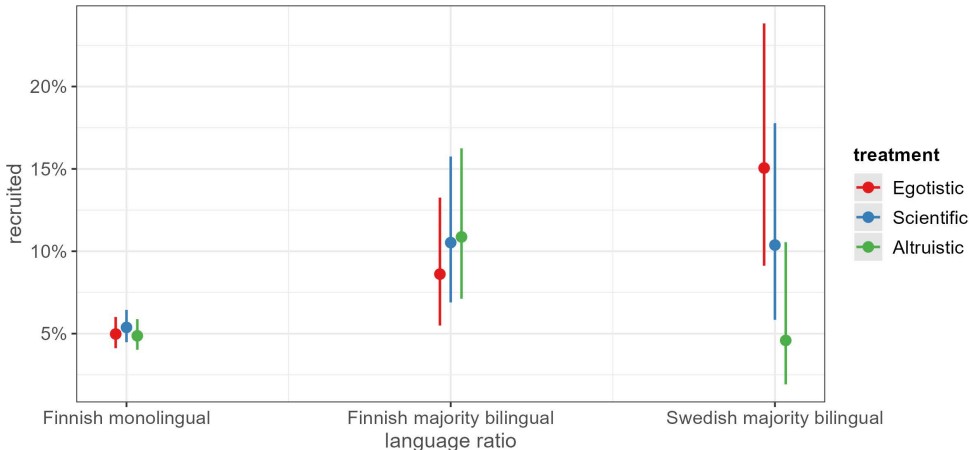

**Fig 5. Interaction municipality language ratio and treatment.**

R-indicators are calculated based on characteristics known for both recruited and non-recruited persons: gender (man, woman), age group (18 34,35 49, 50-64, 65-), language (Finnish, Other), municipality type (urban, suburban, rural), municipality language ratio (Finnish monolingual, Finnish majority bilingual, Swedish majority bilingual), and party type (left, right). R-indicators take values on an interval of 0–1, where 1 is strong representativeness, and 0 is the maximum deviation from strong representativeness [25]. We also calculate 95% confidence intervals to approximate if there are statistically significant differences in R-indicators between the treatment groups or compared to the full sample (Fig 6).

The R-indicators in Fig 6 approach one, suggesting very close similarity between the treatment groups and the target population. There are no statistically significant differences across the groups, which means that acceptance of the invitation between the treatments has resulted in recruited groups that are similar and representative of the target population, regardless of the treatment. This is a reassuring finding because it implies that different appeals do not lead to differences in who complies with the invitation and who does not.

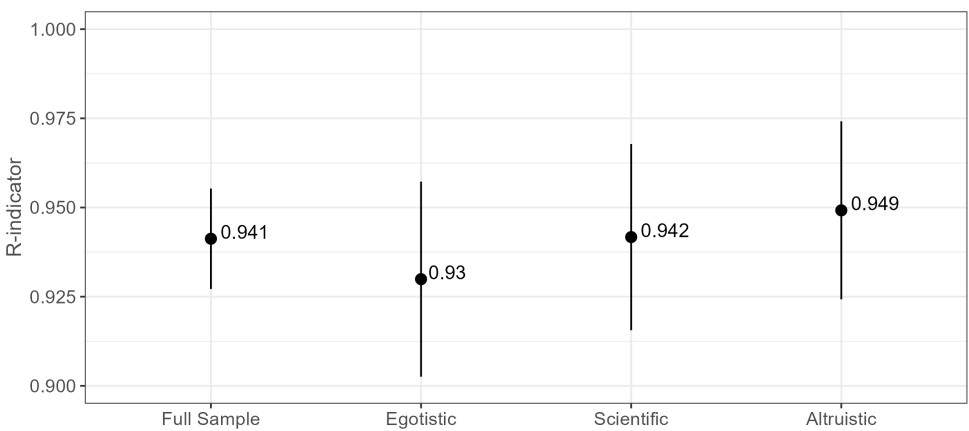

**Fig 6. R-indicators (95% CI) for the full sample and by treatment group.**

## Discussion

Busy politicians are a tough crowd to recruit to scientific survey panels. Against the hypotheses, the findings suggest that, besides some potential effects among certain sub-populations, different invitation appeals do not affect their propensity to accept the invitation. While the null finding is discouraging for survey researchers who would like to increase cooperation rates, the findings are not only disheartening. Appealing to egoistic or altruistic benefits does not lower acceptance rates either, which is important, because it means that the risk of making serious errors in invitation design is low. Also, different appeals lead to the same (and high) representation of the target population. Regardless of framing, the same people accept or turn down the invitation.

While the findings regarding main effects are not substantive, the interactions nevertheless revealed some potential effects, such as Swedish-speaking politicians complying more often than Finnish-speaking politicians, suggesting that the sender affects cooperation rates, because the invitation came from the only Swedish-speaking university in the country. Moreover, a minority position may increase the appeal of the egoistic message. Women demonstrate higher acceptance rates than men, although this does not lead to imbalances in how the target population is represented in the sample. Still, these subgroup analyses should be interpreted with caution, as statistical support is weaker for smaller groups, and successful recruitment, to begin with, was rare.

The null-findings regarding the appeals may be due to several reasons. First, the treatments may not have been strong enough. Being hard-pressed for time and having to divide their attention between different things, politicians perhaps need particularly tough treatment in order to react in any way. In practical terms, however, sending a forceful panel invitation might be counterproductive. The goal of such a message is to be clear and concise, because long e-mails from unknown senders are a certain turn-off for politicians who get similar invites on a daily basis. Thus, the ecological validity of our experiment is arguably high. It was conducted in a real-life setting with the goal of attracting new respondents and the design of the study served this purpose, rather than a hypothesis-testing purpose. Consequently, the null finding is, in some ways, a comforting result for researchers although it undermines the theoretical conception of appeals as a way to enhance survey participation. Second, due to a relatively low acceptance rate, it could be that the experiment did not capture enough behavioral variation among politicians to fully reveal the impact of the treatments.

However, while we chose to approach the effectiveness of invitation wordings among samples of politicians through the leverage-saliency and social conformity frameworks, future scholarship could test wordings informed by prospect theory [26]. Its core claim, that people are more risk averse than benefit-seeking, has received some support in the context of survey recruitment among citizen samples [(see, e.g. 27, 28)]. So, it could be that appeals still work, but just not the ones we tested.

There are also other plausible mechanisms future research could address. One possible approach is to consider politicians' responses to survey invitations through the lens of information overload, which is an important ingredient in the professional lives of busy politicians [4,5]. If the driving force behind politicians' behavior is their need to cope with all the information and requests they receive, short written appeals in e-mails are not a particularly effective recruitment strategy. What could instead work better is ensuring that the survey has maximum visibility in the media, which might incentivize publicity-seeking politicians to participate. Moreover, also the political context of politicians differs significantly, plausibly affecting how they react to survey invitations. Being in a position of power versus in political opposition could affect their sense of altruism and selfishness and, consequently, their willingness to engage in a survey panel. Those in power might have even less capacity to devote to anything outside the political agenda, while those who are relatively powerless might have more time and even a stronger need to be able to voice their opinions any way they can – even through a survey. In sum, different appeals could work depending on the political context and in this analysis, we were unable to account for the variation in municipal-level political dynamics.

Alternatively, the appeals to altruism and egoism could be overridden by pre-existing ideas about a sense of duty to serve the public, including scientific research. In Finland, where research is publicly funded, attitudes towards participating in an academic survey panel could be steered by the more general feelings towards the public sector, or more specifically,

 

the inviting academic institution. Similarly, attitudes towards science in general could largely pre-determine the outcome regardless of appeals and other invitation-related strategies. Politicians do not see surveys as particularly useful sources of information when it comes to public opinion, but instead prefer other sources, such as direct citizen contacts [29]. The hesitancy about surveys and polls in general could also help explain the relative ineffectiveness of the scientific benefit message and the low overall cooperation rate.

In conclusion, future research regarding how to maximize acceptance rates among politicians to participate in surveys should explore two possibilities. First, how can researchers best capture the attention of busy politicians – is it by promising public visibility or by employing appeals that are stronger than those used in the recruitment of citizens? Second, are all attempts to frame the invitation or the study in an appealing way useless, because of pre-existing attitudes towards research and civic duty?

While the findings in this analysis are restricted to a single country and to the local level, the general disposition of politicians in terms of civic duty, patterns of ideological positioning towards science, and risk of informational overload are likely to be similar across other Western democracies. However, the varying political contexts may generate differences, which diminish the generalizability of the findings. Future research should strive to replicate the experiment both in similar and different contexts. Replicating the experiment in a similar context would provide insight into whether the null findings are de facto the result of nonexistent effects or simply a Type II error. Replication in a very different political context would give insight into whether the findings are generalizable also outside of the quite distinct Finnish local-level political system.

## Supporting information

**S1 File.** Online Appendix.
(DOCX)

## Author contributions

**Conceptualization:** Lauri Rapeli, Kim Backström.

**Data curation:** Kim Backström.

**Formal analysis:** Kim Backström.

**Funding acquisition:** Lauri Rapeli.

**Investigation:** Lauri Rapeli.

**Methodology:** Lauri Rapeli, Kim Backström.

**Project administration:** Lauri Rapeli.

**Resources:** Lauri Rapeli.

**Writing – original draft:** Lauri Rapeli, Kim Backström.

**Writing – review & editing:** Lauri Rapeli.

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
