## [Decision Letter · Decision Letter 0]

19 Nov 2025

PONE-D-25-30014Comparison of altruistic, egoistic, and scientific appeals in the recruitment of politicians to a survey panelPLOS ONE

Dear Dr. Rapeli,

Thank you for submitting your manuscript to PLOS ONE. After careful consideration, we feel that it has merit but does not fully meet PLOS ONE’s publication criteria as it currently stands. Therefore, we invite you to submit a revised version of the manuscript that addresses the points raised during the review process.

In particular, all three reviewers see merit in your work but at the same time raise important concerns (especially R1 and R2). Following these constructive comments, **I recommend focusing on clarifying the theoretical aspects of the paper and making the pre-registration material accessible.**
**Plus, including more detail about the experimental treatments and the basic post-experimental checks is essential. Finally, addressing the issue of null findings with better theoretical reasoning is important.**

We look forward to receiving your revised manuscript.

Kind regards,

Cengiz Erisen

Academic Editor

PLOS ONE

**Journal Requirements:**

1. When submitting your revision, we need you to address these additional requirements. Please ensure that your manuscript meets PLOS ONE's style requirements, including those for file naming. The PLOS ONE style templates can be found at https://journals.plos.org/plosone/s/file?id=wjVg/PLOSOne_formatting_sample_main_body.pdf and https://journals.plos.org/plosone/s/file?id=ba62/PLOSOne_formatting_sample_title_authors_affiliations.pdf 2. In your Methods section, please include additional information about your dataset and ensure that you have included a statement specifying whether the collection and analysis method complied with the terms and conditions for the source of the data. 3. Thank you for stating the following financial disclosure: This study has received funding from The Research Council of Finland grants Research 327997 and 345714.   Please state what role the funders took in the study.  If the funders had no role, please state: "The funders had no role in study design, data collection and analysis, decision to publish, or preparation of the manuscript." If this statement is not correct you must amend it as needed. Please include this amended Role of Funder statement in your cover letter; we will change the online submission form on your behalf. 4. Please note that your Data Availability Statement is currently missing the direct link to access each database. If your manuscript is accepted for publication, you will be asked to provide these details on a very short timeline. We therefore suggest that you provide this information now, though we will not hold up the peer review process if you are unable. 5. When completing the data availability statement of the submission form, you indicated that you will make your data available on acceptance. We strongly recommend all authors decide on a data sharing plan before acceptance, as the process can be lengthy and hold up publication timelines. Please note that, though access restrictions are acceptable now, your entire data will need to be made freely accessible if your manuscript is accepted for publication. This policy applies to all data except where public deposition would breach compliance with the protocol approved by your research ethics board. If you are unable to adhere to our open data policy, please kindly revise your statement to explain your reasoning and we will seek the editor's input on an exemption. Please be assured that, once you have provided your new statement, the assessment of your exemption will not hold up the peer review process. 6. Please include your full ethics statement in the ‘Methods’ section of your manuscript file. In your statement, please include the full name of the IRB or ethics committee who approved or waived your study, as well as whether or not you obtained informed written or verbal consent. If consent was waived for your study, please include this information in your statement as well. 7. Please upload a new copy of Figure 2 as the detail is not clear. Please follow the link for more information:  https://journals.plos.org/plosone/s/figures 8. If the reviewer comments include a recommendation to cite specific previously published works, please review and evaluate these publications to determine whether they are relevant and should be cited. There is no requirement to cite these works unless the editor has indicated otherwise.

Reviewers' comments:

Reviewer's Responses to Questions

**Comments to the Author**

1. Is the manuscript technically sound, and do the data support the conclusions?

Reviewer #1: Yes

Reviewer #2: Yes

Reviewer #3: Yes

2. Has the statistical analysis been performed appropriately and rigorously? 

Reviewer #1: Yes

Reviewer #2: Yes

Reviewer #3: Yes

3. Have the authors made all data underlying the findings in their manuscript fully available?

Reviewer #1: Yes

Reviewer #2: No

Reviewer #3: Yes

4. Is the manuscript presented in an intelligible fashion and written in standard English?

Reviewer #1: Yes

Reviewer #2: Yes

Reviewer #3: Yes

5. Review Comments to the Author

**Reviewer #1:** This manuscript presents a large-scale randomized field experiment testing whether altruistic, egoistic, or scientific appeals improve the recruitment of local politicians into a survey panel. Its strengths lie in its novelty—politicians are an especially hard-to-reach group—its methodological rigor, including pre-registration and representativity checks, and its transparency in reporting. The null findings, showing that none of the appeals significantly increased participation, are valuable in themselves, as they challenge expectations derived from citizen-based recruitment studies and underscore the limitations of simple message framing when targeting elites. The analysis also provides useful descriptive insights, showing that age, gender, municipality type, and party affiliation predict participation, and that despite low response rates, the recruited sample closely resembles the broader population of politicians.

That said, the paper would benefit from a clearer theoretical framing and a more cautious interpretation of results. The altruism–egoism distinction is borrowed from citizen survey literature without fully justifying its relevance to politicians, while the scientific appeal is only weakly distinguished from altruism. The discussion of null results is underdeveloped, focusing mainly on practical explanations like information overload rather than deeper theoretical alternatives, such as whether civic duty norms or institutional trust shape politicians’ responsiveness. Subgroup analyses, such as differential effects by municipality type or minority status, are interesting but risk appearing post-hoc and should be presented as exploratory. Finally, the very low response rates raise questions about long-term panel viability and external validity beyond Finland’s high-trust context, which the discussion should address more directly.

Overall, the manuscript makes a useful methodological contribution by showing that framing appeals alone is insufficient to improve participation among politicians. To maximize its impact, the authors should position the study more clearly as a baseline for future work, strengthen the conceptual discussion, and highlight alternative strategies—such as appeals rooted in prospect theory, leveraging trusted institutional endorsements, or exploring non-textual methods of engagement. With these refinements, the paper could make a valuable contribution to both survey methodology and the study of elite participation.

Suggestions:

- Reframe the contribution more explicitly as a methodological demonstration that simple appeal framing does not increase recruitment among politicians, rather than as a theoretical advance in leverage–saliency theory.

- Clarify the theoretical rationale for using altruistic, egoistic, and scientific appeals with elite respondents. More strongly justify why politicians, unlike citizens, should be expected to respond differently to these frames.

- Deepen the discussion of null results by considering alternative explanations, such as institutional trust, civic duty, or elite-specific norms, rather than attributing the outcome primarily to information overload.

- Present subgroup analyses as exploratory rather than confirmatory. This will prevent over-interpretation of interaction effects that lack strong theoretical grounding.

- Discuss the implications of low response rates more directly, including risks of cumulative attrition in panel studies and limits to generalizability beyond the Finnish context.

-Improve readability by streamlining descriptive material and moving some tables and statistical outputs to appendices.

**Reviewer #2:** Thank you for the opportunity to review this paper. The study investigates how different types of motivational appeals affect local politicians’ willingness to participate in a research panel. Using a large-scale experimental design with more than 7,000 local-level politicians, the paper makes a comprehensive attempt to understand to extent to which participation in such surveys may be incentivized. The authors should be commended for assembling a rich dataset and addressing an important methodological question. However, several aspects of the design, analysis, and presentation could be improved before the manuscript is ready for publication. Some of these issues are fairly major in nature, which, in my opinion, might require extensive revisions. My comments are in no particular order.

Starting with the theory, the assumption that an egoistic appeal works because politicians value access to the panel’s findings is strong and should be treated more cautiously. A growing body of elite-survey research suggests that many politicians do not perceive direct benefits from public-opinion information, nor do they believe such information meaningfully advance their careers. Acknowledging this in the discussion section would help readers understand why the treatment may not have produced significant differences (i.e., perhaps because the stimulus was not strong enough to generate a behavioral response).

A related point concerns the likely diversity in what motivates politicians. Different forms of egoism and altruism may appeal to different types of politicians, depending on institutional, electoral, and media contexts. It would be useful for the authors to acknowledge that such cross-context and role-based variation could partly explain the absence of clear treatment effects.

The paper refers to three experimental groups repeatedly before clearly specifying what these groups are. Introducing the treatment conditions earlier in the paper would provide an overview of the design to the reader and make the design easier to follow.

The second hypothesis is under-theorized relative to the first and third. Strengthening the theoretical justification here, perhaps by linking it to distinct mechanisms or prior findings, would improve coherence across the hypotheses. There was a brief mention about left-wing politics, but I haven't seen much about past research detailing the mechanism behind this.

I was unclear about the pre-registration process. The text refers to a pre-registration (“see the pre-registration for more details”) but then suggests that it was shared only with the editors. I was not able to identify a separate submission document that details the pre-registration process. Reviewers know who the authors are, so I don't understand why the authors preferred not to reveal it. If the document is really available only to the editors, it prevents reviewers from assessing deviations from the original design, which undermines the purpose of pre-registration. Clarifying this and, ideally, sharing the pre-registration document as supplementary material would increase transparency and credibility.

With over 7,000 politicians in the sample, the study is large enough to examine heterogeneous treatment effects. It would strengthen the contribution to explore whether the treatment effects differ across subgroups. Even descriptive exploration of heterogeneity would enrich the interpretation of otherwise null or weak average effects.

The discussion of low acceptance rates across treatment groups suggests that bias from rare events could affect the estimates. Replicating the main models using a penalized maximum-likelihood estimator, such as Firth’s logistic regression for rare events (see King & Zeng; Allison), would be a valuable robustness check.

The manuscript lacks a clear methodological section explaining why specific estimators were chosen and how balance was assessed. Most experimental studies report randomization checks or balance tests; this paper does not clearly do so. Table A1 includes significance tests, but it is unclear what they really tested, and many are statistically significant. A simple regression predicting treatment assignment with covariates would demonstrate whether randomization succeeded. There are many other tables in the appendix, but it is difficult to understand what they do. A detailed discussion accompanying those tables would be very helpful. The authors should walk readers through these diagnostics explicitly.

Assuming randomization was successful, control variables may not be necessary. Presenting models without controls would help assess the robustness of the findings and avoid potential post-treatment bias. In other words, comparing simple and full specifications would increase confidence in the results.

Overall, I believe there is enough promise in this study for eventual publication, but significant revisions are necessary as there are major issues in the paper. I encourage the authors to view these comments as constructive steps toward strengthening an already ambitious and important project.

**Reviewer #3:** The paper represents methodologically rigorous experimental design, and has transparent reporting and strong adherence to reproducibility standards. It addresses an important methodological gap in survey recruitment of political elites such as politicians. It also has clear practical and theoretical contributions to survey methodology and political behavior literature.

6. PLOS authors have the option to publish the peer review history of their article (what does this mean?). If published, this will include your full peer review and any attached files.

Reviewer #1: No

Reviewer #2: **Yes:** Tevfik Murat Yildirim

Reviewer #3: No

---

## [Author Response · Author response to Decision Letter 1]

31 Jan 2026

We thank for the opportunity to revise and resubmit our manuscript. We really appreciated all the comments and have done our best to address them. Below, we respond to each of them in the order as they appear in the decision letter. We look forward to hearing your views on the revised manuscript.

It should be noted that we have rerun all analyses, and excluded the previous party name variable, and for the language variable, merged those with Swedish and other mother tongues. This is done to ensure anonymity for the individuals in the data, and enable the sharing of the data and analysis code.

REVIEWER 1:

Reframe the contribution more explicitly as a methodological demonstration that simple appeal framing does not increase recruitment among politicians, rather than as a theoretical advance in leverage–saliency theory.

Yes, we agree that this is the right choice for this paper. We are happy to see that the reviewers have understood and appreciated the aim of our paper, which is to contribute to the methodology of recruiting politicians and where the focus is very pragmatic rather than theory-heavy. In the Introduction, we now write as follows, in order to explain the pragmatic aim of the study:

We study the impact of invitation framing for pragmatic reasons. In their role as elected policymakers who serve the interests of the public, accepting – or expecting – money for advancing research on democracy and perhaps their own political accountability does not seem suitable. This is particularly the case in many countries, such as Finland, where research is primarily publicly funded. Instead of monetary incentives, which may be feasible for attracting ordinary citizens to participate in surveys but not equally appropriate when approaching democratically elected politicians, it is necessary to investigate other ways to improve recruitment success, such as whether different framings could affect politicians’ willingness to participate in a survey panel. We examine the possibility that politicians could be persuaded by appealing to their altruistic sense of responsibility to contribute to research about how democracy works due to their position as elected representatives of the people. Alternatively, a more egoistic appeal suggests that by becoming a panel participant, they are offered access to key findings from the panel, which are highly relevant to their work in local politics. If effective, simple appeals to participate would be an easy and cost-free method for survey researchers to increase cooperation rates in politician samples.

Clarify the theoretical rationale for using altruistic, egoistic, and scientific appeals with elite respondents. More strongly justify why politicians, unlike citizens, should be expected to respond differently to these frames.

This is a fair point. Although maybe arguing for a difference between politicians and citizens is not the main point of the analysis (given that we make no such comparison), we acknowledge that we had not motivated the rationale for using the appeals very well. Admittedly, if there was no reason to think that politicians and citizens would in way react differently, there perhaps would not be any reason to focus specifically on politicians.

We added the following to the beginning of the section that explains the use of the different appeals. As we argue, politicians are likely to have a stronger sense of civic duty, which could make them more likely to react to the appeals.

The practical issue is how can an invitation be tailored so that it has enough saliency for politicians to cooperate? On the one hand, we assume that politicians have a strong sense of civic duty. Altruism and a sense of civic duty are strong predictors of political engagement (9,10) and local-level politicians are extraordinarily highly engaged individuals. Additionally, research has shown that a motivating factor in becoming a politician in the first place is civic duty, i.e. contributing (altruistically) to society (11). On the other hand, politicians compete for attention to survive in electoral competition. Consequently, it also seems plausible that politicians’ reactions to survey invitations could be affected by more selfish considerations about how to make themselves heard, rather than simply be considered an anonymous data entry (see also 5).

Deepen the discussion of null results by considering alternative explanations, such as institutional trust, civic duty, or elite-specific norms, rather than attributing the outcome primarily to information overload.

As also Reviewer #2 suggested, alternative explanations could still be added. In addition to information overload and weakness of the treatment(s), in this revised version we also discuss sense of civic duty, attitudes towards science and the impact of political context as plausible reasons for why we did not find effects beyond some sub-groups.

Present subgroup analyses as exploratory rather than confirmatory. This will prevent over-interpretation of interaction effects that lack strong theoretical grounding.

We agree with the reviewer that especially some of our heterogeneous treatment effect analyses should be viewed as exploratory. We now elaborate on this in the methods section and also conduct descriptive analyses as a complement to the interactions.

See the Appendix for a closer description of the included variables. Our rationale for including the variables is to improve precision in identifying treatment effects, detect potential biases, and enable us to study heterogeneous treatment effects. As mentioned when discussing the hypotheses, we believe party type affiliation plays a role in the effectiveness of the treatments. Regarding other individual-level demographic variables, we include age due to it regularly being an important predictor of (non-)response in surveys (Lundmark & Backström, 2025). Gender and language are relevant due to potential differences in treatment effects, due to perceptions of being in the majority or minority locally or nationally. Women are generally underrepresented in politics, meaning they may perceive the treatments differently from men, and likewise, the effect of the treatments may differ for those with a mother tongue other than Finnish. The context-level variables, while potentially suffering from ecological validity issues, may also provide insights into the context in which the politicians operate and potential differences in the effects of the treatments. There may be differences in resources and time budgets between municipalities of differing urbanicity, while a bilingual context can also have implications for representativeness, e.g., due to differences in political culture. Still, we view our investigation of these potential heterogenous effects as purely explorative.

[...]

We also examine heterogeneous treatment effects by comparing recruitment rates across different subpopulations. See Figure A1 and Table A3 in the Appendix. We can note some general trends where women, older, and leftist politicians in general are more prone to join the panel. However, there are no significant differences between treatments within the different groups. Still, while marginally insignificant, the egotistic treatment performed somewhat better among politicians in suburban municipalities. Likewise, the egotistic treatment performed somewhat better, although not statistically significant, among politicians in Swedish-majority bilingual municipalities.

Discuss the implications of low response rates more directly, including risks of cumulative attrition in panel studies and limits to generalizability beyond the Finnish context.

Cumulative attrition, i.e. attrition in a panel over the course of several waves, is indeed a key issue for any panel, along with low response rates. In this paper, we have mentioned low response rates as one reason why successful recruitment is so important. However, we are hesitant about entering the discussion on response rates and cumulative attrition, because we view them as adjacent questions and we do not seek a contribution to the (large) bodies of literature on these subjects.

Nevertheless, generalizability is always an issue with single-country studies. We suggest adding the following to the end of the Discussion:

While the findings in this analysis are restricted to a single country, the general disposition of politicians in terms of civic duty, patterns of ideological positioning towards science and risk of informational overload, are likely to be similar across other Western democracies. However, the varying political contexts may generate differences, which diminish the generalizability of the findings.

Improve readability by streamlining descriptive material and moving some tables and statistical outputs to appendices.

We understand that tables and figures may affect the readability of an article. At this point, we suggest moving Table 1 to the Appendix, which we have done. It is now Table A3. If the reviewer or the editor feels more removals are needed, Figures 4 and 5 could be included in the Appendix as well, because they are somewhat explorative. Nevertheless, we felt they belong to the storyline of the paper and for the time being kept them in the body text.

REVIEWER 2:

The assumption that an egoistic appeal works because politicians value access to the panel’s findings is strong and should be treated more cautiously. A growing body of elite-survey research suggests that many politicians do not perceive direct benefits from public-opinion information, nor do they believe such information meaningfully advance their careers. Acknowledging this in the discussion section would help readers understand why the treatment may not have produced significant differences (i.e., perhaps because the stimulus was not strong enough to generate a behavioral response).

It is also our interpretation that among some other likely reasons, the treatment was not strong enough to provoke a reaction. We agree with the reviewer that suggesting that the egoistic appeal would work due to politicians valuing access to the panel’s findings sounds unlikely. In the initial submission, we wrote this in one sentence in the introduction, where we explained the rationale behind the treatments. Now, based on various reviewer comments, we have instead emphasized the actual treatment text, which read “By participating in the panel, you will be able to voice your opinion about democracy in Finnish municipalities and about different societal issues.” Thus, the expected impact would be due to politicians’ desire to make their own opinions heard, not access to the panel’s findings. Moreover, the reviewer is correct in pointing out that politicians likely do not consider survey data as being very useful or relevant to their work. In our analysis, this is probably best reflected in the fact the scientific altruistic message was mostly ineffective.

We made two changes. First, we removed the sentence suggesting that politicians would participate because of access to the panel’s findings. Second, we agree with the reviewer that it would be useful to comment in the Discussion on politicians’ attitudes toward opinion data, as one possible reason for the lack of treatment effects. With a reference to Walgrave & Soontjens (2023), we suggest the following addition:

Politicians do not see surveys as particularly useful sources of information when it comes to public opinion, but instead prefer other sources, such as direct citizen contacts (22). The hesitancy about surveys and polls in general could also help explain the relative ineffectiveness of the scientific benefit message and the low overall cooperation rate.

Different forms of egoism and altruism may appeal to different types of politicians, depending on institutional, electoral, and media contexts. It would be useful for the authors to acknowledge that such cross-context and role-based variation could partly explain the absence of clear treatment effects.

This is a very good point. We added the following to the Discussion:

Moreover, also the political context of politicians differs significantly, plausibly affecting how they react to survey invitations. Being in a position of power versus in political opposition could affect their sense of altruism and selfishness and, consequently, their willingness to engage in a survey panel. Those in power might have even less capacity to devote to anything outside the political agenda, while those who are relatively powerless might have more time and even a stronger need to be able to voice their opinions any way they can – even through a survey. In sum, different appeals could work depending on the political context and in this analysis, we were unable to account for the variation in municipal-level political dynamics.

The paper refers to three experimental groups repeatedly before clearly specifying what these groups are. Introducing the treatment conditions earlier in the paper would provide an overview of the design to the reader and make the design easier to follow.

This is a good point and simple to fix. We now explain the treatments already in the introduction where they are mentioned for the first time.

The second hypothesis is under-theorized relative to the first and third. Strengthening the theoretical justification here, perhaps by linking it to distinct mechanisms or prior findings, would improve coherence across the hypotheses. There was a brief mention about left-wing politics, but I haven't seen much about past research detailing the mechanism behind this.

We readily admit that H2 was too superficially justified. We rewrote the justification, with reference to https://doi-org.ezproxy.utu.fi:2443/10.1177/07311214211022391 and http://dx.doi.org/10.1146/annurev-soc-030320-035037. Including new references to previous research, it now reads as follows:

In addition to a general altruistic appeal, we include a more specific version called the scientific appeal. It emphasizes the importance of volunteering as a panel member to contribute to the advancement of science. In line with the saliency theory, the scientific treatment message expects to increase the sense of relevance of participating, which could be particularly effective among politicians who supposedly wish to contribute to the common good. However, the saliency of scientific benefits of participating could plausibly depend on ideological leaning. In recent years, climate change and Covid-19 have become strongly contentious issues in terms of attitudes towards science and conservatives display significantly less trust in science than liberals and/or left-leaners (16–18). Distrust in science is also associated with behavioral outcomes, such as lower support for science-based policy and for scientifically grounded recommendations (16). Consequently, politicians in the political left could more likely be persuaded by the scientific appeal, because they tend to have more faith in science and also behave accordingly.

I was unclear about the pre-registration process. The text refers to a pre-registration (“see the pre-registration for more details”) but then suggests that it was shared only with the editors. I was not able to identify a separate submission document that details the pre-registration process. Reviewers know who the authors are, so I don't understand why the authors preferred not to reveal it. If the document is really available only to the editors, it prevents reviewers from assessing deviations from the original design, which undermines the purpose of pre-registration. Clarifying this and, ideally, sharing the pre-registration document as supplementary material would increase transparency and credibility.

We had somehow simply missed that the reviewers will get access to our names already at this point and therefore retracted the link to the pre-registration from the manuscript, which is of course the praxis with most journals. The link is https://doi.org/10.17605/OSF.IO/C3T4X and it is now also included in the manuscript. Our apologies for the confusion.

With over 7,000 politicians in the sample, the study is large enough to examine heterogeneous treatment effects. It would strengthen the contribution to explore w

---

## [Decision Letter · Decision Letter 1]

9 Mar 2026

PONE-D-25-30014R1Comparison of altruistic, egoistic, and scientific appeals in the recruitment of politicians to a survey panelPLOS One

Dear Dr. Rapeli,

Thank you for submitting your manuscript to PLOS ONE. After careful consideration, we feel that it has merit but does not fully meet PLOS ONE’s publication criteria as it currently stands. Therefore, we invite you to submit a revised version of the manuscript that addresses the points raised during the review process. In particular, R2 raises a few important concerns that require your attention. I believe these are easily doable within a reasonable timeframe. Please submit your revised manuscript by Apr 23 2026 11:59PM. If you will need more time than this to complete your revisions, please reply to this message or contact the journal office at plosone@plos.org. Please include the following items when submitting your revised manuscript:

We look forward to receiving your revised manuscript.

Kind regards,

Cengiz Erisen

Academic Editor

PLOS One

Journal Requirements:

Reviewers' comments:

Reviewer's Responses to Questions

**Comments to the Author**

1. If the authors have adequately addressed your comments raised in a previous round of review and you feel that this manuscript is now acceptable for publication, you may indicate that here to bypass the “Comments to the Author” section, enter your conflict of interest statement in the “Confidential to Editor” section, and submit your "Accept" recommendation.

Reviewer #1: All comments have been addressed

Reviewer #2: All comments have been addressed

2. Is the manuscript technically sound, and do the data support the conclusions?

Reviewer #1: Yes

Reviewer #2: Yes

3. Has the statistical analysis been performed appropriately and rigorously? 

Reviewer #1: Yes

Reviewer #2: Yes

4. Have the authors made all data underlying the findings in their manuscript fully available?

Reviewer #1: Yes

Reviewer #2: Yes

5. Is the manuscript presented in an intelligible fashion and written in standard English?

Reviewer #1: Yes

Reviewer #2: Yes

6. Review Comments to the Author

Reviewer #1: The manuscript examines whether different invitation framings influence politicians’ willingness to join a survey panel. Using a large field experiment among more than 7,000 local politicians in Finland, the authors compare altruistic, egoistic, and scientific appeals in recruitment emails. The study addresses an important methodological problem in elite research: how to increase participation rates in surveys targeting political elites. The experimental design and the use of real behavioral outcomes represent clear strengths of the study.

The research design is generally sound. The use of randomized assignment and a large sample size provides a solid basis for causal inference. The statistical analyses are broadly appropriate for the research question. The authors employ logistic regression models and also conduct robustness checks using a penalized likelihood estimator suitable for rare events, which increases confidence in the stability of the results. The addition of randomization checks and alternative model specifications also improves transparency. Overall, the statistical approach is adequate and supports the main conclusions.

The paper’s main finding that simple framing strategies do not significantly increase recruitment among politicians is interesting and potentially valuable for scholars conducting elite surveys. Reporting null results from well-designed field experiments can make an important contribution, particularly in areas where practical methodological guidance is limited. The discussion appropriately acknowledges several possible explanations for the lack of treatment effects, including weak treatments, information overload, and broader contextual factors affecting politicians’ willingness to participate in surveys.

The authors have also responded constructively to the main points raised in the previous round of review. In particular, the manuscript now frames its contribution more clearly as a methodological demonstration rather than a strong theoretical test, and additional robustness checks and diagnostic information have been included. These revisions improve the transparency of the analytical strategy and clarify the scope of the study’s contribution.

That said, several aspects of the manuscript could still be strengthened. First, the theoretical justification for the three framing strategies remains somewhat limited. While the authors refer to civic duty and self-interest as potential motivations for politicians, the mechanisms linking these motivations to the specific treatments could be more clearly articulated. Strengthening this theoretical discussion would help clarify why different framing strategies were expected to generate different behavioral responses.

Second, the heterogeneous analyses are potentially interesting but appear to be largely exploratory. The manuscript would benefit from emphasizing this point more clearly and avoiding overly strong interpretations of subgroup patterns, particularly when statistical support is weak. Presenting these analyses as descriptive extensions rather than confirmatory tests would improve analytical transparency.

Third, although the authors discuss limitations related to generalizability, this issue could be elaborated further. Because the study focuses on local politicians in a single national context, it is not clear to what extent the findings apply to other political systems or to politicians operating at different institutional levels.

Finally, some elements of the methodological description could be presented more clearly in the main text rather than primarily in the appendix. In particular, a brief explanation of the randomization checks, model selection strategy, and robustness analyses would improve readability and make the analytical strategy easier to follow.

Overall, the manuscript presents a well-designed field experiment addressing an important methodological issue in elite research. With clearer theoretical framing and some improvements in the presentation of the analysis and interpretation of subgroup results, the paper could provide a useful contribution to the literature on elite survey methodology.

Reviewer #2: I thank the authors for their revisions. My comments have been sufficiently addressed and I am happy to recommend publication of the manuscript.

7. PLOS authors have the option to publish the peer review history of their article (what does this mean?). If published, this will include your full peer review and any attached files.

Reviewer #1: **Yes:** Emre Erdogan

Reviewer #2: **Yes:** Tevfik Murat Yildirim

---

## [Author Response · Author response to Decision Letter 2]

25 Mar 2026

Response letter

Dear editor and reviewers

Thank you for the opportunity to review and resubmit our paper “Comparison of altruistic, egoistic, and scientific appeals in the recruitment of politicians to a survey panel”.

We have now revised the paper and have submitted a version both with and without tracked changes. Also, we have now submitted replication data and code, as well as figures in TIFF format (300 dpi).

Here follows a detailed response to Reviewer 1’s concerns.

1. Strengthen the theoretical justification for the three treatments

Response: The reviewer would still like to see a better theoretical discussion of the mechanism behind the rationale for the treatments. We have added social conformity theory as the fundamental explanation for why we expect that politicians could respond to our treatments.

In addition to some smaller changes elsewhere, the theory section now includes the following:

The practical issue is how can an invitation be tailored so that it has enough saliency for politicians to cooperate? Fundamentally, we draw on the central tenet of social conformity theory, according to which people’s behaviors and actions depend on what they assume is expected of them because of their social positions and role in any situation (see e.g. ,9,10). Thus, performing in their role as an elected politician when they receive the invitation, we expect the politicians to be persuaded by appeals that refer to what is perhaps societally expected behavior in this particular situation. On that note, we assume that politicians may feel that demonstrating have a strong sense of civic duty is something that conforms to social expectations. Additionally, research has shown that altruism and a sense of civic duty are strong predictors of political engagement (11,12) and local-level politicians are extraordinarily highly engaged individuals. Additionally, research has also shown that a motivating factor in becoming a politician in the first place is civic duty, i.e. contributing (altruistically) to society (13).

2. Make it clearer that the subgroup analyses are exploratory

Response: Throughout the paper, we have eased up the wording regarding the heterogeneous analyses, making it clearer that they should be viewed as exploratory and that their statistical power is more limited.

Namely:

In Figures 4 and 5, we highlight statistically significant findings outside of the hypotheses simply as an exploratory approach to identify potential underlying effects that are otherwise hidden by the main effects.

[…]

Against the hypotheses, the findings suggest that, besides some potential effects among certain sub-populations, different invitation appeals do not affect their propensity to accept the invitation.

[…]

The interactions nevertheless revealed some potential effects

[…]

Still, these subgroup analyses should be interpreted with caution, as statistical support is weaker for smaller groups, and successful recruitment, to begin with, was quite rare.

3. Discuss limitations regarding generalizability, both regarding the Finnish and local-level cases.

Response: We have now elaborated in the final paragraph of the conclusions regarding this:

While the findings in this analysis are restricted to a single country and at the local political level

[…]

Future research should strive to replicate the experiment both in similar and different contexts. Replicating the experiment in a similar context would provide insight into whether the null findings are de facto the result of nonexistent effects or simply a Type II error. Replication in a very different political context would give insight into whether the findings are generalizable also outside of the quite distinct Finnish local-level political system.

4. Move methodological considerations into the main text

Response: We have now deleted the two footnotes on randomization checks and model selection and have instead included more detailed descriptions in the main text.

When discussing the randomization, we have added:

As a randomization check, we compare the sample compositions of the three treatment groups concerning gender, age, language, municipality type, municipality language ratio, and political party classification. We find no statistically significant differences between the groups, indicating successful randomization (see Table A1).

When discussing the predicted probabilities, we have added:

We also use logistic regression and predicted probabilities to examine predictors of recruitment. We test four different models and compare their fit (see Appendix Table A5 and Table A6. The chosen model (model 4) had the best fit, and we further examined its quality using the performance-package (Lüdecke et al., 2019). We conclude that the model had a good fit, although it showed some issues with binned residuals, suggesting a slight risk of overpredicting recruitment for some groups (see Figure A2). Across all four models, the overall treatment effects are not significant. As a robustness check, we also tried running the same model using Firth’s logistic regression, which can be more suitable when working with unbalanced data (20). However, we note no substantive differences between the regular MLE logistic regression and Firth’s logistic regression (Table A7 in the Appendix), meaning we continue using the MLE logistic regression model.

---

## [Decision Letter · Decision Letter 2]

12 Apr 2026

PONE-D-25-30014R2Comparison of altruistic, egoistic, and scientific appeals in the recruitment of politicians to a survey panelPLOS One

Dear Dr. Rapeli,

Thank you for submitting your manuscript to PLOS ONE. After careful consideration, we feel that it has merit but does not fully meet PLOS ONE’s publication criteria as it currently stands. Therefore, we invite you to submit a revised version of the manuscript that addresses the points raised during the review process. Please submit your revised manuscript by May 27 2026 11:59PM. If you will need more time than this to complete your revisions, please reply to this message or contact the journal office at plosone@plos.org. Please include the following items when submitting your revised manuscript:

We look forward to receiving your revised manuscript.

Kind regards,

Cengiz Erisen

Academic Editor

PLOS One

Journal Requirements:

Additional Editor Comments:

R1 appreciates the second round of revisions but requests one final set of comments. I kindly request addressing them as soon as possible, after which I will reach an in-house decision at the earliest.

Reviewers' comments:

Reviewer's Responses to Questions

**Comments to the Author**

1. If the authors have adequately addressed your comments raised in a previous round of review and you feel that this manuscript is now acceptable for publication, you may indicate that here to bypass the “Comments to the Author” section, enter your conflict of interest statement in the “Confidential to Editor” section, and submit your "Accept" recommendation.

Reviewer #1: (No Response)

2. Is the manuscript technically sound, and do the data support the conclusions?

Reviewer #1: Yes

3. Has the statistical analysis been performed appropriately and rigorously? 

Reviewer #1: Yes

4. Have the authors made all data underlying the findings in their manuscript fully available?

Reviewer #1: Yes

5. Is the manuscript presented in an intelligible fashion and written in standard English?

Reviewer #1: Yes

6. Review Comments to the Author

Reviewer #1: Thank you for the opportunity to review the revised version of your manuscript. The paper addresses an important and underexplored issue, how to recruit political elites into survey panels—and does so with a transparent and well-executed experimental design. The revisions have improved the manuscript in several meaningful ways, particularly in terms of clarity and scope. At the same time, a few core issues remain only partially resolved.

First, the theoretical framework has been strengthened with the inclusion of social conformity theory. This addition provides a clearer rationale for why politicians, as role holders, might respond to normative cues embedded in survey invitations. However, the link between this framework and the specific treatments (altruistic, egoistic, scientific appeals) remains somewhat underdeveloped. The argument now clarifies why normative expectations may matter, but it is still not fully clear why these particular framings should generate differential effects. As a result, the theoretical contribution would benefit from a more explicit articulation of the mechanisms connecting role-based expectations to the specific experimental manipulations.

Second, the handling of subgroup analyses is notably improved. The manuscript now consistently frames these analyses as exploratory and acknowledges their limited statistical power. This represents a clear gain in interpretive caution. That said, parts of the discussion still attribute substantive meaning to marginal or statistically weak patterns. I would encourage further restraint here, ensuring that these findings are presented strictly as hypothesis-generating.

Third, the discussion of generalizability is much stronger. The added reflections on the Finnish local-level context and the explicit call for replication in other settings appropriately bound the scope of the findings. This revision directly addresses earlier concerns and improves the manuscript’s external validity claims.

Fourth, moving key methodological details into the main text significantly enhances transparency. The inclusion of randomization checks, model comparisons, and robustness analyses makes the empirical strategy easier to evaluate and strengthens confidence in the results.

Two issues, however, remain insufficiently addressed.

The first concerns the interpretation of the null findings. While the manuscript now acknowledges uncertainty, it still leans toward concluding that framing does not matter for elite recruitment. Given the very low baseline participation rate, it is plausible that behavioral variation is constrained, limiting the observable impact of treatments. This alternative explanation limited elasticity rather than absence of effect should be more directly incorporated into the interpretation.

The second concerns broader theoretical integration. The discussion introduces several plausible mechanisms, such as information overload, institutional trust, and reputational considerations, but these remain post hoc explanations rather than analytically integrated alternatives. The manuscript would be stronger if it more explicitly positioned these mechanisms alongside the tested framework, even if only as directions for future research.

7. PLOS authors have the option to publish the peer review history of their article (what does this mean?). If published, this will include your full peer review and any attached files.

Reviewer #1: No

---

## [Author Response · Author response to Decision Letter 3]

22 Apr 2026

Thank you for the review comments and for the opportunity to revise and resubmit once more. Below, we respond to the remaining comments in the order as they appear in the decision letter. We look forward to hearing what you think of the revised version of our manuscript.

1) The theoretical contribution would benefit from a more explicit articulation of the mechanisms connecting role-based expectations to the specific experimental manipulations.

While the reviewer appreciated the revised theoretical section, something seems to still be lacking in the way it connects with the experimental design. Although the we and the reviewer may have slightly different understandings of what constitutes a ‘mechanism’ in this type of research, we have done our best to articulate even more explicitly what effects we expected from the design. As we understand it, a ‘mechanism’ in this context can only be a reaction in the respondent’s mind, prompted by words in the actual panel invitation. To some extent, our design is exploratory, because we simply do not know what kind of an appeal is going to work best. Hence, we expect the same basic ‘mechanism’ to be at work in all instances, which is very simply that appealing to a person’s sense of duty will make them more likely to comply with a request. To be very clear on this, we added the following:

Regarding the altruistic treatments: Therefore, the mechanism, which we expect to trigger a response in the respondents as they read the invitation with the altruistic appeals, is the feeling that politicians’ are expected to display a sense of duty towards the good of society.

Regarding the egoistic treatment: In this case, the mechanism assumed to be at play is the more self-centered need to seek visibility and focus, rather than the need to conform to duty-based role expectations.

2) Parts of the discussion still attribute substantive meaning to marginal or statistically weak patterns. I would encourage further restraint here, ensuring that these findings are presented strictly as hypothesis-generating.

We understand that the reviewer wants us to be even more cautious about the interpretation of the findings than what we currently are. In our own reading of how we present the findings, we mostly just say that the appeals we used did not seem to work, rather than make strong claims about statistically weak connections (if this is what the reviewer means). For example, we begin the discussion by referring to ‘potential effects among some sub-populations’. Additionally, we conclude that ‘Still, these subgroup analyses should be interpreted with caution, as statistical support is weaker for smaller groups, and successful recruitment, to begin with, was rare’. These conclusions, in our view, do not attribute significant substantive meaning to the findings. However, we have revisited this section once more and made small adjustments, which further emphasize the uncertainty of the findings. This also includes our response to the next reviewer comment.

3) Given the very low baseline participation rate, it is plausible that behavioral variation is constrained, limiting the observable impact of treatments. This alternative explanation limited elasticity rather than absence of effect should be more directly incorporated into the interpretation.

The reviewer makes a valid and useful point with this comment, which we see as related to the previous one. We added the following to the discussion where we comment on the null findings:

Second, due to a relatively low acceptance rate, it could be that the experiment did not capture enough behavioral variation among politicians to fully reveal the impact of the treatments.

4) The discussion introduces several plausible mechanisms, such as information overload, institutional trust, and reputational considerations, but these remain post hoc explanations rather than analytically integrated alternatives. The manuscript would be stronger if it more explicitly positioned these mechanisms alongside the tested framework, even if only as directions for future research.

Our aim has always been to present these as plausible mechanisms to be tested by future research, rather than ‘post hoc explanations’. After all, we cannot address these possibilities empirically with our (pre-registered) design anyway, so they cannot really be anything else but directions for subsequent analyses. With small adjustments in the discussion, we have now framed them more clearly as precisely that – hopefully to avoid any further confusion.

---

## [Editor Report · Decision Letter 3]

27 Apr 2026

Comparison of altruistic, egoistic, and scientific appeals in the recruitment of politicians to a survey panel

PONE-D-25-30014R3

Dear Dr. Rapeli,

We’re pleased to inform you that your manuscript has been judged scientifically suitable for publication and will be formally accepted for publication once it meets all outstanding technical requirements.

Kind regards,

Cengiz Erisen

Academic Editor

PLOS One
---

## [Editor Report · Acceptance letter]

PONE-D-25-30014R3

PLOS One

Dear Dr. Rapeli,

I'm pleased to inform you that your manuscript has been deemed suitable for publication in PLOS One. Congratulations! Your manuscript is now being handed over to our production team.

Kind regards,

on behalf of

Dr. Cengiz Erisen

Academic Editor

PLOS One